# CR-MoE: Consistent Routed Mixture-of-Experts for Scaling Contrastive Learning

**Ziyu Jiang**                                                                *jiangziyu@tamu.edu*
*Texas A&M University*

**Guoqing Zheng**                                                           *zheng@microsoft.com*
*Microsoft Research*

**Yu Cheng**                                                          *chengyu@cse.cuhk.edu.hk*
*The Chinese University of Hong Kong*

**Ahmed Hassan Awadallah**                                        *hassanam@microsoft.com*
*Microsoft Research*

**Zhangyang Wang**                                                        *atlaswang@utexas.edu*
*University of Texas at Austin*

**Reviewed on OpenReview:** *https://openreview.net/forum?id=qKIvn9xL1R*

## Abstract

While Contrastive Learning (CL) achieves great success in many downstream tasks, its good performance heavily relies on a large model capacity. As previous methods focus on scaling dense models, training and inference costs increase rapidly with model sizes, leading to large resource consumption. In this paper, we explore CL with an efficient scaling method, Mixture of Experts (MoE), to obtain a large but sparse model. We start by plugging in the state-of-the-art CL method to MoE. However, this naive combination fails to visibly improve performance despite a much larger capacity. A closer look reveals that the naive MoE+CL model has a strong tendency to route two augmented views of the same image token to different subsets of experts: such "cross-view instability" breaks the weight-sharing nature in CL and misleads the invariant feature learning. To address this issue, we introduce a new regularization mechanism, by enforcing expert-routing similarity between different views of the same image (or its overlapped patch tokens), while promoting expert-routing diversity of patches from different images. The resultant method, called CR-MoE, improves by 1.7 points in terms of 1% semi-supervised learning accuracy on ImageNet, compared to the naive combination baseline. It further surpasses the state-of-the-art CL methods on ImageNet pre-training of Vision Transformer (ViT) by 2.8 points, at the same computational cost. Our findings validate CR-MoE as an effective and efficient image representation learner. Code is available at `https://github.com/VITA-Group/CRMoE`.

## 1 Introduction

Unsupervised contrastive Learning (CL) has been popularly explored as it demonstrate strong performance on many downstream tasks, which could even beat its supervised counterpart (Chen et al., 2020c; Grill et al., 2020; Caron et al., 2020; Chen et al., 2021b; Caron et al., 2021). However, the performance of CL heavily relays on the large capacity of the employed model. For instance, in semi-supervised learning with few labels, one important application of self-supervised learning (Tian et al., 2020b), SimCLR-v2 (Chen et al., 2020b)

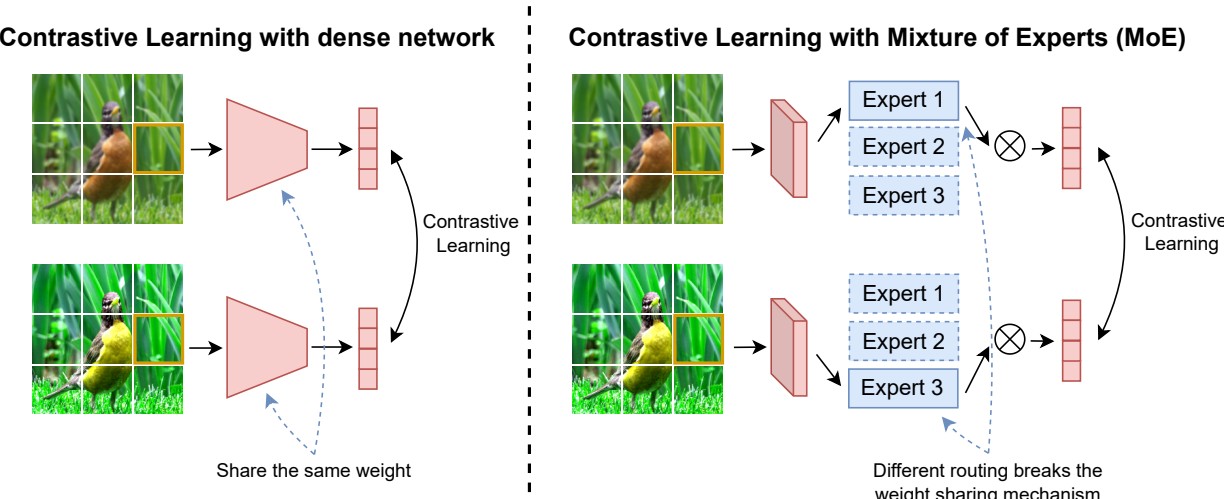

Figure 1: Routing comparison between the traditional dense network (left) and naive MoE+CL (right). In contrastive learning of the dense network, two branches always share the same weights. However, naively adopting MoE to CL can lead to different routing predictions for the same patch and break the weight-sharing mechanism.

demonstrates that scaling model parameters from 24M to 795M brings a performance improvement by 17%. However, scaling dense models significantly increases the training and inference cost. For instance, The 795M model would increase the training time by 41 times, and training to full performance (1000 epochs) on ImageNet-1K (Deng et al., 2009) requires 7000 GPU (V100) days.

In this paper, we study employing an efficient scaling method, a sparse Mixture of Experts (MoE) (Shazeer et al., 2017), for CL, without sacrificing training and inference efficiency. In contrast to dense models that process each sample with all parameters, MoE leverages a dynamic sparse model: each sample is routed to a small subset of experts. And each expert is a small Muti-Layer Perceptron (MLP) network. In this way, a large candidate pools of experts can be built while only activating a small part for each sample, making it possible to leverage large model capacity while maintaining small computational costs for training and inference. MoE has been applied successfully in NLP applications (Lepikhin et al., 2020; Fedus et al., 2021) and was recently introduced to vision tasks but only for supervised settings (Riquelme et al., 2021).

We start with directly applying CL on vision MoE models (e.g. Riquelme et al. (2021)). However, we find this naive combination only yields marginal performance improvement compared to its dense counterpart despite a much larger capacity. Looking closer, we observe that different augmented views of the same image tokens are mostly routed to different subsets of experts (as illustrated in Figure 1), This essentially breaks the conventional design of contrasting **shared weight branches** (Chen et al., 2020a; He et al., 2020; Grill et al., 2020) and turns to contrasting **independent branches**, which we show hurts performance with further empirical evaluations.

To enforce consistency in expert selections for augmented image views, a naive way is to always assign them the same set of experts. However, this leaks the learning target of CL: the instance identity, causing the model to overfit on such trivial nuisance without learning meaningful image representations (Chen et al., 2021a). Instead, as shown in Figure 2, we propose a simple yet effective regularization mechanism to enforce the consistency of expert selection based on visual overlapping. Specifically, first we pair all image tokens based on the overlapping between patches. Then we pull the selection of experts of paired tokens to be similar while differentiating that for tokens from different images through the proposed Overlapping-based Gate Aligning Regularization (OGAR). The resulting method, termed CR-MoE, significantly improves the consistency of the experts selection for different augments of the same image and the 1% semi-supervised

performance by 1.7 points compared to the naive plugin, which is also 2.8 points higher than competing state-of-the-art CL methods on ViT.

Our contributions are summarized as follows:

- We propose CR-MoE, which efficiently scales Contrastive Learning (CL) with the sparse Mixture of Experts, pushing the limit of CL towards large model capacity while maintaining similar computation cost.
- We identify the problem of naively combining MoE and CL, which essentially routes semantically similar images to different sets of experts thus hurting performance, and address it by proposing a novel regularization loss.
- Extensive experiments verifies the effectiveness of the proposed regularization term. Compared to competitive state-of-the-art CL methods on ViT, the proposed CR-MoE achieves an improvement of 2.8 points at the same computational cost.

## 2 Related works

### 2.1 Self-supervised training

Inspired by the observation that conducting instance recognition could yield a good representation that naturally clusters the same class images (Alexey et al., 2016; Wu et al., 2018), various works devote to designing self-supervised learning through pulling the representations of the same images together while pushing those of different images apart (Chen et al., 2020c;a; He et al., 2020; Tian et al., 2020a), also known as contrastive learning. Some works also recognize that negative samples are not necessary (Grill et al., 2020; Misra & Maaten, 2020; Chen & He, 2021; Zbontar et al., 2021). A trend was observed and verified by (Chen et al., 2020a;b) that contrastive learning yields better performance with a longer training schedule and a large backbone model. However, training large models with CL for a long schedule imposes significantly high training costs. In this work, to scale CL we investigate an efficient scaling option based on Mixture-of-Experts(MoE). While recent work (Meng et al., 2022) also starts to explore sparsifying the contrastive learning with dynamic pruning strategies, MoE has its unique strength on memory efficiency and combining it with contrastive learning is still not explored.

Other works on self-supervised learning focus on the handcrafted pretext tasks (Trinh et al., 2019) like rotation prediction (Gidaris et al., 2018), jigsaw (Noroozi & Favaro, 2016; Carlucci et al., 2019) and colorization (Gidaris et al., 2018). Recent advances in transformer highlight the possibility of a new class of self-supervised learning methods through masked image modeling (Bao et al., 2021; He et al., 2021b; Xie et al., 2021). These conceptually different directions can also be combined with contrastive learning to further boosting the performance (Dangovski et al., 2021; Zhou et al., 2021). In this work, we focus on studying contrastive learning while leaving other directions as potential future work.

### 2.2 Sparse Mixture of Experts

The traditional Mixture of Experts Network is composed of multiple sub-models and conduct input conditional computation (Jacobs et al., 1991; Jordan & Jacobs, 1994; Chen et al., 1999; Yuksel et al., 2012; Roller et al., 2021). While contrastive learning can also be improved with the traditional MoE (Tsai et al., 2020), it suffers from intensive computation since the model are dense and all experts are activated. Recent work (Shazeer et al., 2017) proposes the Sparse Mixture of Experts Layer and demonstrates better results on language modeling with lower computational cost. Following works devise methods to further address the communication cost (Fedus et al., 2021; Lewis et al., 2021) and stability (Zoph et al., 2022) issues. GLaM (Du et al., 2021) studies the MoE for language self-supervised task and achieve significant downstream few-shot performance.

MoE is recently applied for computer vision tasks (Riquelme et al., 2021; Gross et al., 2017; Xue et al., 2021; Wang et al., 2020; Tsai et al., 2018; Ahmed et al., 2016; Yang et al., 2019; Pavlitskaya et al., 2020). However, most of these works focus only on supervised or weakly supervised learning. Recently, LIMoE (Mustafa et al.,

2022) starts to explore applying MoE on self-supervised language-image pairing tasks, where they propose a local and global entropy design to balance different modalities. In this work, we reveal and address the challenge from inconsistent expert routing when applying MoE to self-supervised vision tasks.

## 3 Method

### 3.1 Preliminaries

**Contrastive learning** Contrastive learning is a self-supervised method via maximizing instance discriminativeness. For example, it enforces the similarity of positive pairs while enlarging the distance of negative pairs (Wu et al., 2018):

$$\mathcal{M}(v_i, v_i^+, V^-, \tau) = -\frac{1}{N} \sum_{i=1}^{N} \log \frac{s_\tau(v_i, v_i^+)}{s_\tau\left(v_i, v_i^+\right) + \sum_{v_i^- \in V^-} s_\tau(v_i, v_i^-)} \tag{1}$$

where $v_i^+$ is considered a positive sample of sample $v_i$ while the set $V^-$ consists of negative samples. $s_\tau(v_i, v_i^+) = \exp\left(v_i \cdot v_i^+ / \tau\right)$ measures the similarity of positive pair $(v_i, v_i^+)$ while $s_\tau\left(v_i, v_i^-\right)$ measure the similarity of negative pair $(v_i, v_i^-)$. $\tau$ is the temperature controlling the magnitude of all terms.

MoCo-v3 (Chen et al., 2021b) is one of the state-of-the-art self-supervised methods devised for ViT (Dosovitskiy et al., 2020). It encodes two crops $C_1$ and $C_2$ for each image under random data augmentation. The images are then encoded with network and its Exponential Moving Average (EMA). MoCo-v3 also introduce random token projection to stabilize the learning process. The loss of MoCo-v3 is defined as

$$L_{\text{CL}} = \mathcal{M}(f_1, f_2, \{f\}^-, \tau) = -\frac{1}{N} \sum_{i=1}^{N} \log \frac{s_\tau(f_1, f_2)}{s_\tau\left(f_1, f_2\right) + \sum_{f^- \in \{f\}^-} s_\tau(f_1, f^-)} \tag{2}$$

where the features $(f_1, f_2)$ encoded from $(C_1, C_2)$, respectively, are employed as positive samples while negative set $\{f\}^-$ is composed by the features of views from other images.

**Sparse Mixture of Experts** MoE reduces the computational cost via activating a small subset of computational graph for each sample. The basic building block of MoE is the sparse MoE layers, which consists of $n_e$ expert networks $(E_1, E_2, \cdots, E_{n_e})$. Formally, a MoE layer is defined as

$$y = \sum_{i=1}^{n_e} G(x)_i E_i(x) \tag{3}$$

where $x$ and $y$ are the input and output, respectively. $G$ is the gating function that outputs a vector containing scores for each expert network $E_i(x)$, typically instantiated with a Softmax. By picking the top-$k$ scored experts ($k << n_e$), the model only activates a small subset of expert networks for each sample. For $G$, we employ the noisy top-k gating design introduced in Riquelme et al. (2021) as

$$G(x) = \text{TopK}(\text{Softmax}(Wx + \epsilon), k) \tag{4}$$

where $W$ is a learnable weight and $\epsilon$ denotes Gaussian noise sampled from $\mathcal{N}\left(0, \frac{1}{n_e^2}\right)$. $Wx$ controls the clean score of the gating function while noise in $\epsilon$ benefits the load balancing between experts. The sum of the score are then normalized with Softmax function and sparsified with TopK defined as

$$\text{TopK}(v, k)_i = \begin{cases} v_i & \text{if } v_i \text{ is in the top } k \text{ elements of } v \\ 0 & \text{otherwise.} \end{cases} \tag{5}$$

In this work, we focus on studying applying MoE for the ViT (Dosovitskiy et al., 2020) backbone. We follow the strategy of Riquelme et al. (2021) to replace every other multi-layer perceptron (MLP) layers with sparse

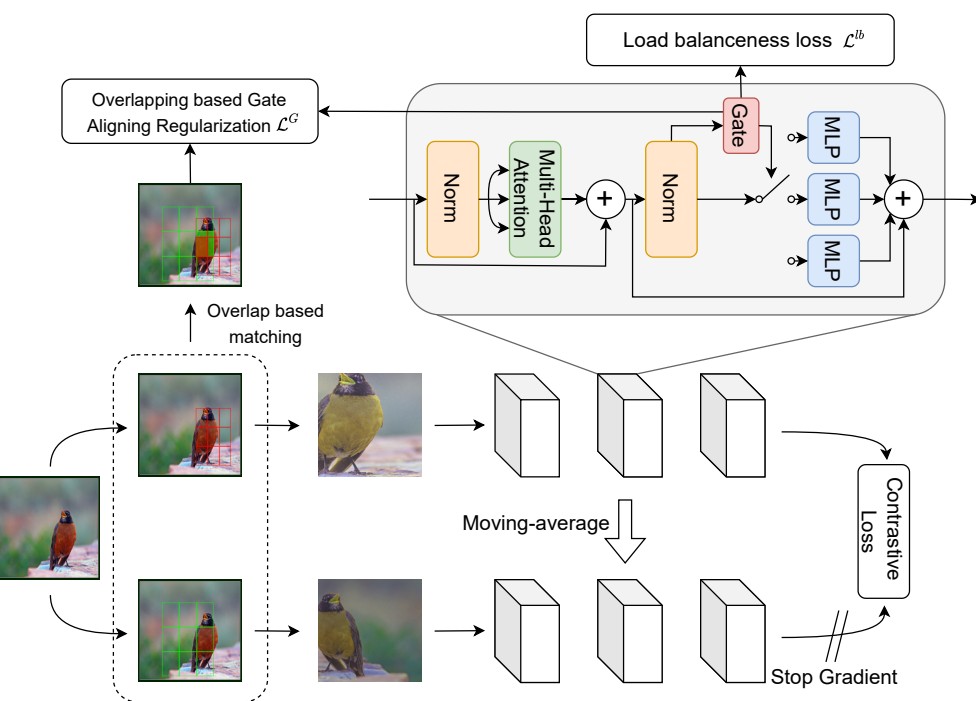

Figure 2: Pipeline of the proposed CR-MoE. It replaces every other block of ViT to sparse MoE layer. Overlapping based gate aligning regularization is applied for training the proposed network.

**MoE layers.** Each expert network is of the same architecture: $\mathrm{MLP}(x) = W_2\sigma_{\mathrm{gelu}}(W_1 x)$, where $W_1 \in \mathbb{R}^{d_m \times d_f}$ and $W_2 \in \mathbb{R}^{d_f \times d_m}$ are learanble weights while $\sigma_{\mathrm{gelu}}$ is the non-linear activation layer (Hendrycks & Gimpel, 2016). It is worth noting that MoE is applied to multiple visual tokens, where each token could have different expert choice.

We also employ an auxiliary loss to encourage the load balancedness following Shazeer et al. (2017) termed as $\mathcal{L}^{\mathrm{lb}}$ to prevent the over-selection of few experts.

### 3.2 Sparse Mixture of Experts for Contrastive Learning

To enforce the consistency of expert selection while not leaking image identity, we introduce a new regularization term called Overlapping based Gate Aligning Regularization (OGAR). In ViT, MoE layer would choose experts for each token. The token sequence includes one classification token and multiple patch tokens. We then introduce how OGAR is applied for classification and patch tokens.

**OGAR for classification tokens** As the classification token is at the image level, enforcing consistency can be easily realized by applying the similarity constraint among classification tokens for augments of the same image. Formally, it is defined as

$$
\begin{aligned}
\mathcal{L}_{[\mathrm{CLS}]}^{G} &= \mathcal{M}(G_{[\mathrm{CLS}]}^{1}, G_{[\mathrm{CLS}]}^{2}, \{G_{[\mathrm{CLS}]}\}^{-}, \tau) \\
&= -\frac{1}{N}\sum_{i=1}^{N}\log \frac{s_\tau(G_{[\mathrm{CLS}]}^{1}, G_{[\mathrm{CLS}]}^{2})}{s_\tau\left(G_{[\mathrm{CLS}]}^{1}, G_{[\mathrm{CLS}]}^{2}\right) + \sum_{G_{[\mathrm{CLS}]}^{-} \in \{G_{[\mathrm{CLS}]}\}^{-}} s_\tau(G_{[\mathrm{CLS}]}^{1}, G_{[\mathrm{CLS}]}^{-})}
\end{aligned}
\tag{6}
$$

where $G_{[\mathrm{CLS}]}^{1}$ and $G_{[\mathrm{CLS}]}^{2}$ denote the gating function output ($G(x)$ in Equation 4) of classification tokens from a pair of positive samples. $\{G_{[\mathrm{CLS}]}\}^{-}$ denotes those from negative samples. $\tau$ is the temperature, where we use the same value as $\mathcal{L}_{\mathrm{CL}}$. We employ the form of Moco V3 loss to enforce the consistency for preventing all the gate functions collapse to always outputting the same prediction.

**OGAR for patch tokens** Unlike classification tokens, different patches lack one-to-one correspondence as the patches are randomly sampled from different regions of the original image. Hence matching the patches is required before conducting the regularization. Previous studies reveal that the transformer can automatically learn object segmentation that aligns well with input in terms of the spatial location (Caron et al., 2021), which indicates the strong spatial correlation between input and features learned by CL. Inspired by this observation, we design a matching method based on the spatial location of the patches. As shown in Figure 2, each patch of one view is paired with the most overlapping patch from the other view. For one patch that is not overlapped enough with any other patches (below a certain overlapping threshold $\lambda$), we leave it unpaired. Only those paired patches are utilized for calculating the loss. Formally, the proposed loss on patch $p_m$ is defined as

$$\mathcal{L}_{p_m}^G = \begin{cases} -\frac{1}{N} \sum_{i=1}^{N} \log \frac{s_\tau(G_m, G_n)}{s_\tau(G_m, G_n) + \sum_{G^- \in \{G\}^-} s_\tau(G_m, G^-)} & \text{if } \text{IoU}_{mn} > \lambda \\ 0 & \text{otherwise.} \end{cases}, \quad n = \arg\max_{n'} \text{IoU}_{mn'} \quad (7)$$

where $p_n$ denotes the patch that has the largest Intersection over Union (IoU) with $p_m$. $\text{IoU}_{mn}$ represents the IoU between patch $m$ and $n$. $G_m$ and $G_n$ are gating function outputs for $p_m$ and $p_n$, respectively. $\{G\}^-$ denotes those from negative patch samples. When the $\text{IoU}_{mn}$ is less than threshold $\lambda$, the loss is 0. Otherwise, the consistency loss between $G_m$ and $G_n$ would be employed. The overall gating loss is averaged over all patches as

$$\mathcal{L}_p^G = \frac{1}{N_p} \sum_{m=1}^{N_p} \mathcal{L}_{p_m}^G \quad (8)$$

where $N_p$ is the number of patches.

Some previous works study a similar problem: enforcing the regional regularization of CL (Li et al., 2021; Wang et al., 2021), which also requires matching the local features. They match the features across two views based on the feature distance (e.g. cosine similarity). However, we empirically find this approach yield less significant improvement in our case. The intuition behind this is that the paired features in inter-mediate layers may lack strong feature similarity. The proposed matching method allows the existence of non-paired patches while the design of (Wang et al., 2021) assumes all local features can be paired, which is prone to noise in learned features and also in general does not hold in practice.

We balance the two regularization terms with a convex combination controlled with a weight $\alpha$ ($0 < \alpha < 1$). Formally, the resulting OGAR is

$$\mathcal{L}^G = (1 - \alpha)\mathcal{L}_{[\text{CLS}]}^G + \alpha\mathcal{L}_p^G \quad (9)$$

**The overall optimization target for CR-MoE** To sum up, the overall loss is

$$\mathcal{L} = \mathcal{L}_{\text{CL}} + w_{\text{lb}}\mathcal{L}^{\text{lb}} + w_G\mathcal{L}^G \quad (10)$$

where $w_{\text{lb}}$ and $w_G$ are the scaling factor of the loading balancedness losses and OGAR, respectively. By employing OGAR on naive MoE+CL, the resultant CR-MoE framework can efficiently scaling contrastive learning with MoE.

## 4 Experiment

### 4.1 Settings

**Pre-training** Our pre-training experiments are conducted on ImageNet-1K (Deng et al., 2009) following common practice (Chen et al., 2020a; He et al., 2020). For pre-training framework, we employ Moco v3 (Chen et al., 2021b), and we follow the same settings as Moco v3 on data augmentations and learning specification: 3-layer MLP projection head, temperature $\tau = 0.2$, momentum $m = 0.99$, random patch projection, cosine decay schedule (Loshchilov & Hutter, 2016), and 40-epoch warmup. For optimization, we

employ AdamW (Loshchilov & Hutter, 2017) optimizer and a weight decay of 0.1. We employ linear scaling rule (Goyal et al., 2017) and search for the best base learning rate (lr) on 100-epoch results with grid of $\{1.5e^{-4}, 3.0e^{-4}, 5.0e^{-4}, 1.0e^{-3}\}$. The best searched lr is $5.0e^{-4} \times$ BatchSize/256. For model ablations, we employ a shorter schedule of 100 epochs with a relatively small batch size of 1024. When comparing with state-of-the-art methods, we scale up and employ 300 epochs with a batch size of 3072.

**Linear probing** Linear probing measures the quality of learned representations from pre-training. After self-supervised pre-training, we remove the MLP heads and train a classifier with the frozen backbone. Following Moco V3, we employ the SGD optimizer with a batch size of 4096 and weight decay of 0 for 90 epochs, with only random resized cropping and flipping augmentation. The lr is swept following common practice (Chen et al., 2021b; Zhou et al., 2021).

**Semi-supervised and transfer few-shot learning** Learning with few labels is an important application for contrastive learning, which pertains to both semi-supervised and transfer few-shot learning (Chen et al., 2020b; Tian et al., 2020b;b; Islam et al., 2021). Specifically, for semi-supervised learning, we consider 1% or 10% available labels (following the sampling in Chen et al. (2020b)) of ImageNet. For transfer few-shot learning, we consider 4-shot and 10-shot settings for three datasets: CIFAR10 (Krizhevsky et al., 2009), Pet37 (Parkhi et al., 2012) and Food101 (Bossard et al., 2014).

For these two applications, we consider a two steps paradigm: The model is first pre-trained on the *pre-train* and then it is *supervised fine-tune* on the seed or few-shot dataset. For the *supervised fine-tune* step, we employ different settings for different tasks. As suggested in Tian et al. (2020b); Zhou et al. (2021), we train a linear classifier on frozen features for ImageNet 1% semi-supervised task and all transfer few-shot tasks. We optimize for 800 epochs with batchsize of 256 while other settings keeps the same as *linear probing*. For ImageNet 10% semi-supervised task, we follow Chen et al. (2020b); Zhou et al. (2021) fine-tuning from the first layer of the MLP head. The epochs number is set as 200 while the lr are searched with grid of $\{1e^{-5}, 3e^{-5}, 1e^{-4}, 3e^{-4}\}$.

**Hyper-parameters for Mixture-of-Experts Model and loss** For MoE network, we by default employ 16 expert candidates ($n_e = 16$) and always activate 2 of them ($k = 2$). For the employed loss terms, we employ $\lambda = 0.2$, $\alpha = 0.3$, $w_{\text{lb}} = 0.01$ and $w_G = 0.001$, which are searched on 100-epoch training.

For each expert network, we choose $d_f = 2d_m$ instead of $d_f = 4d_m$ in Chen et al. (2021b) to keep the computational cost of activating 2 experts the same as that in ViT. The employed model is VMoE-S/16, as shown in table 1, its FLOPs are comparable to ViT-S/16. Moreover, we further compare the training and inference computation costs in terms of GPU time cost. For inference of a single image on one A6000 GPU,

| Model | Parameters | FLOPs |
|---|---|---|
| ResNet50 | 25M | 4.1G |
| ViT-S/16 | 22M | 4.6G |
| VMoE-S/16 | 72M | 4.6G |
| ViT-B/16 | 87M | 17.6G |

Table 1: Network architecture comparison for four different architectures. CR-MoE uses VMoE-S/16 as the backbone.

the time costs are 1.25ms and 1.07ms for VMoE-S/16 and ViT/S-16, respectively. For training a batch of 1024 images on 8 A6000 GPUs, the time costs are 1.579s and 1.425s for VMoE-S/16 and ViT/S-16, respectively. VMoE-S/16 is only marginally slower than ViT-S/16 in both cases.

**Computation Framework** Our implementation is based on Pytorch (Paszke et al., 2019) and Fast-MoE (He et al., 2021a) library. Models are pre-trained on 32 Nvidia V100 GPUs.

## 4.2 Naive Combination of MoE and CL Does Not Work

In this section, we look into the "cross-view instability" issue of directly plugging MoE to CL and show how the proposed regularization address this problem.

**The routing is inconsistent** To check the consistency of the expert decision, as shown in Figure 3a, we exclude random cropping and flipping from data augmentations to ensure we can locate the different views of the same patches: they are always in the same position in this way. Further, we define these patches with the same content as *corresponding tokens* while defining the tokens from other images as the *non-corresponding*

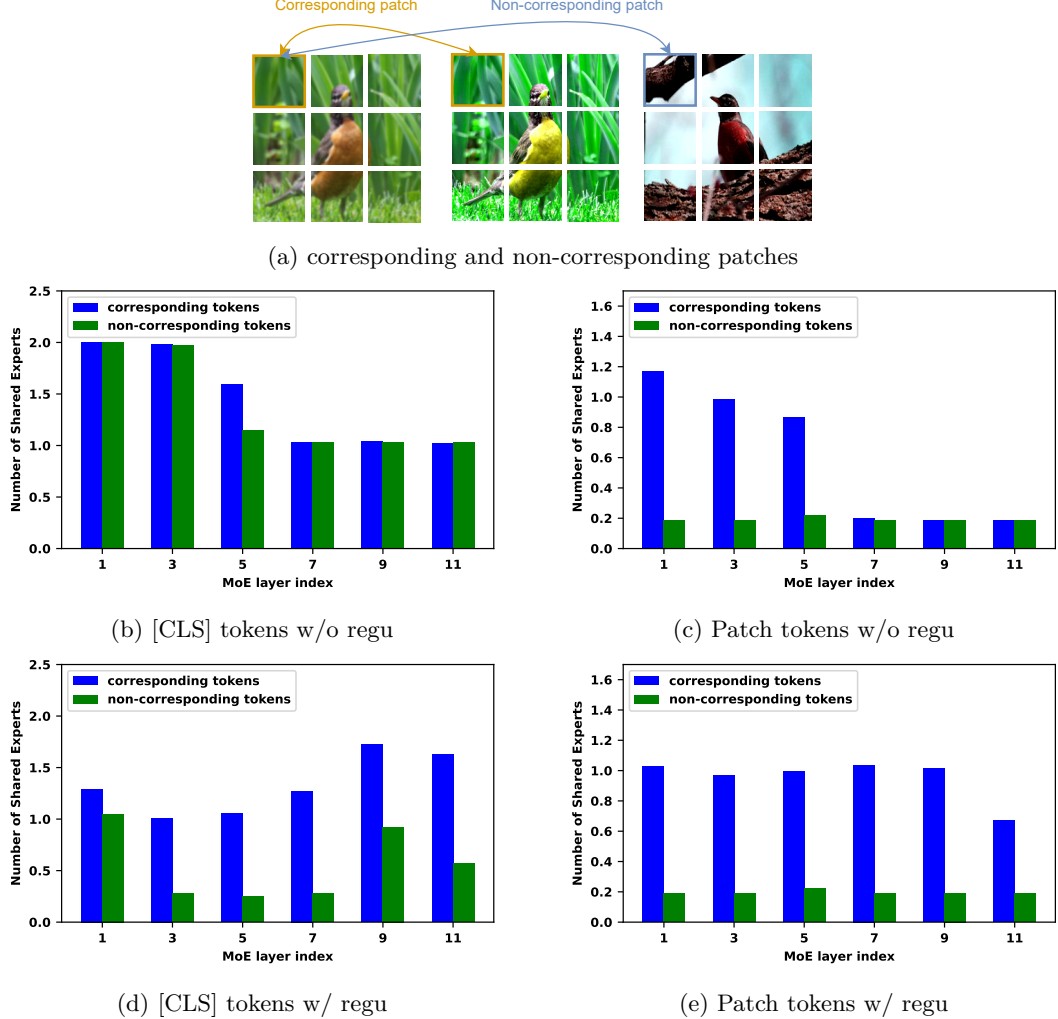

Figure 3: (a) Illustration for the definition of corresponding and non-corresponding patches. The rest four figures compare the average number of shared experts for $G(x)$ between corresponding and non-corresponding tokens. (b)(d) shows the number of shared experts for classification tokens while (c)(e) represents the number of shared experts for patch tokens. All of them are measured across different layers. The "w/o regu" in (b)(c) denotes they are from the naive combination CL and MoE model. In contrast, (d)(e) is from the proposed CR-MoE. The x-axis of the last four figures is the index of the MoE layer in VMoE.

*tokens.* Then, we calculate the average number of shared experts (the number of experts selected by both tokens in the pair) for *corresponding tokens* and *non-corresponding tokens* and make a comparison.

As shown in Figure 3b, for classification tokens of the naive combination, the gating function always selects the similar number of shared experts between *corresponding* and *non-corresponding patches*, which means the difference between corresponding patches and non-corresponding patches can hardly be distinguished. For the patch tokens, as presented in Figure 3c, the boundary between *corresponding* and *non-corresponding patches* get blurred in the deep layers. This would change the standard contrasting shared weight backbone fashion of CL to contrasting (partially) non-shared weight contrastive learning.[1]

**Inconsistent routing leads performance dropping** Unfortunately, the proof-of-concept experiments verify that performance of (partially) non-shared weight contrastive learning can drop. Specifically, we

---

[1]Contrasting with a moving average network can be regarded as sharing weight as the moving average would converge to the online value when training stabilizes.

Table 2: Linear probing (denote as linear) and 1% imagenet semi-supervised (denote as 1%) performance comparison for pre-training and evaluation on ImageNet. All the reported accuracy is top 1 accuracy (%). expert 0/1 for sep-ViT-S denote the two different paths of sep-ViT-S.

| Method | Model | Linear | 1% |
|---|---|---|---|
| Moco v3 | ViT-S/16 | 69.7 | 53.5 |
| Moco v3 | sep-ViT-S/16 (expert 0) | 68.4 | 48.5 |
| Moco v3 | sep-ViT-S/16 (expert 1) | 68.5 | 48.7 |
| Moco v3 | V-MoE-S/16 | 69.9 | 54.1 |
| CR-MoE (Ours) | V-MoE-S/16 | **70.7** | **55.8** |

Table 3: Comparison with State-of-The-Art methods in terms of linear probing (denote as Linear), 1% and 10% semi-supervised performance (denote as 1% and 10%, respectively). All the reported accuracy is top 1 accuracy (%). The SD denotes self-distillation.

| Match method | Model | Linear | 1% | 10% |
|---|---|---|---|---|
| SimCLR v2 (Chen et al., 2020b) | Resnet50 | 71.7 | 57.9 | 68.1 |
| SimCLR v2 + SD (Chen et al., 2020b) | Resnet50 | 71.7 | 60.0 | 70.5 |
| Moco v3 (Chen et al., 2021b) | ViT-S/16 | 73.4 | 59.4 | 72.2 |
| CR-MoE (ours) | V-MoE-S/16 | **74.1** | **62.2** | **73.0** |

designed a special network called sep-ViT, which has the same backbone architecture as MoE with 2 expert candidates. For routing, we would activate different experts for different branches. In this way, these two branches would not share weight in the expert network. The result is illustrated in Table 2, the sep-ViT-S (expert 0) decrease the performance by 0.8% and 5% for linear probing and 1% semi-supervised performance compared to the baseline, respectively, indicating that (partially) non-shared weight can hurt the performance for CL (especially for the semi-supervised performance).

**The proposed CR-MoE improves both consistency and performance** After employing the proposed classification alignment and OGAR, as shown in Figure 3d and 3e, the proposed CR-MoE successfully increase the number of shared experts for *corresponding* tokens while reducing or keeping the number of the shared experts for *non-corresponding tokens*. Also, as shown in Table 2, in contrast to the naive combination of CL and MoE that only improves the baseline Moco V3 by a small margin of 0.2% and 0.6% in terms of linear probing and 1% semi-supervised performance, the proposed CR-MoE increase this margin to 1.0% and 2.3%, demonstrating the effectiveness of the proposed method.

### 4.3 Comparison with State-of-The-Art Methods

In this section, we compare the proposed CR-MoE with state-of-the-art methods. For a fair comparison, we employ a longer training schedule of 300 epochs following Chen et al. (2021b); Caron et al. (2021); Zhou et al. (2021).

Table 4: Transfer few-shot performance comparison across different datasets between MocoV3 and the proposed CR-MoE with ViT-S/16 and V-MoE-S/16, respectively. 4-shot and 10-shot denote 4 and 10 samples available for each class for downstream tasks, respectively. All the reported accuracy is top 1 accuracy (%).

| Dataset | Method | 4-shot | 10-shot |
|---|---|---|---|
| CIFAR10 | Moco V3 | 72.9 | 80.1 |
| | CR-MoE | 74.4 | 80.7 |
| Pet37 | Moco V3 | 71.8 | 81.4 |
| | CR-MoE | 74.4 | 84.3 |
| Food101 | Moco V3 | 35.2 | 48.8 |
| | CR-MoE | 37.4 | 50.1 |

**CR-MoE yield better in-domain performance** As shown in Table 3, the proposed CR-MoE achieves highest performance in terms of Linear probing, 1% and 10% semi-supervised learning. Remarkably, compared to Moco v3 on ViT-S/16, the proposed CR-MoE significantly improves the 1% semi-supervised per-

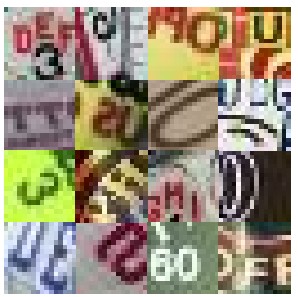 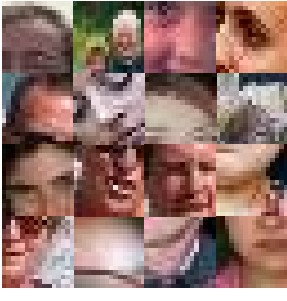 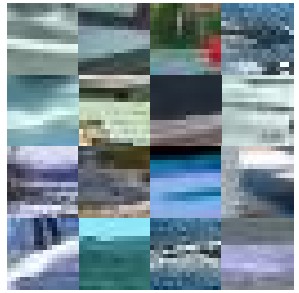 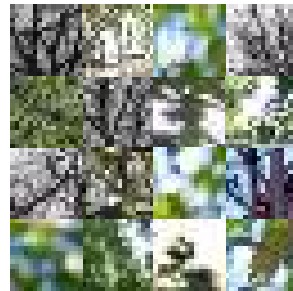

(a) Expert 1 (Characters)  (b) Expert 2 (Faces)  (c) Expert 4 (Pool/Sea) (d) Expert 5 (Forest/Tree)

Figure 4: Visualization of the patch tokens routed to different experts in the 7th layer of CR-MoE on ImageNet. The patches with different patterns are routed to different experts.

formance by 2.8%. Meanwhile, there is also a non-trivial improvement on linear evaluation and 10% semi-supervised performance by 0.6% and 0.8%, respectively. Since the Moco V3 and CR-MoE share the same CL framework, this demonstrate the effectiveness of MoE framework and the proposed regularization. The large improvement on semi-supervised performance also matches the observation at Chen et al. (2020b) that large capacity helps more for few-shot learning.

**CR-MoE yields better transfer few-shot performance** We then study if the strong in-domain few-shot performance can transfer to downstream datasets. As demonstrated in Table 4, the proposed CR-MoE also yields a consistent improvement of [1.5%,0.6%], [2.6%,2.9%] and [2.2%,1.3%] for CIFAR10, Pet37 and Food101, respectively, in terms of [4-shot, 10-shot] performance, demonstrating the proposed CR-MoE can also significantly improve the downstream few-shot performance.

### 4.4 Ablation Studies

**Visualization of routing choices** Following LIMoE (Mustafa et al., 2022), we visualize the routing distribution of CR-MoE in Figure 4. Even though no semantic caption or labels are involved in the training, we find that the patches routed to different tokens show distinct semantic patterns. For example, the patches of [Characters, Faces, Pool/Sea, Forest/Tree] are routed to Expert [1, 2, 4, 5], respectively.

**OGAR loss for patch tokens matters** As shown in Table 5, when removing OGAR loss for patch tokens by setting $\alpha = 0$, the linear evaluation and 1% semi-supervised performance would drop by [0.1%, 0.4%], which demonstrate the effectiveness of the proposed OGAR loss for patch tokens. Other hyper-parameter changes like $w_G = 0.01$ and $\lambda = 0.1$ only marginally change the performance.

Table 5: Comparison between different hyper-parameters settings of the proposed CR-MoE. Linear probing (denote as linear) and 1% imagenet semi-supervised (denote as 1%) performance are reported. All the reported accuracy is top 1 accuracy (%). The first row denotes the employed hyper-parameter setting. Error bar is calculated by running 3 times with different random seeds.

| $w_G$ | $\alpha$ | $\lambda$ | Linear | 1% |
|---|---|---|---|---|
| 0.001 | 0.3 | 0.2 | 70.7±0.07 | 55.8±0.25 |
| 0.001 | 0.0 | 0.2 | 70.6±0.13 | 55.4±0.14 |
| 0.01 | 0.3 | 0.2 | 70.5 | 55.8 |
| 0.001 | 0.3 | 0.1 | 70.6 | 55.9 |

Table 6: OGAR loss ablation regarding different matching methods and whether to employ negative samples. FSM denotes the Feature Similarity-based Matching method employed in Li et al. (2021); Wang et al. (2021).

| Matching Method | Negative samples | Linear | 1% |
|---|---|---|---|
| FSM | ✓ | 70.5±0.07 | 55.6±0.37 |
| Overlap | | 70.2 | 54.2 |
| Overlap | ✓ | 70.7±0.07 | 55.8±0.25 |

In Table 6, we ablation study the proposed OGAR loss. When discarding the negative samples for OGAR loss and only enforcing consistency as in Grill et al. (2020), we observe the gating function tent to choose the same experts for all samples even though we have employed the loading balance loss. Meanwhile, the performance would largely decrease, showing that negative samples are necessary for OGAR. When switching from the overlap-based matching method to the Feature Similarity-based Matching method (FSM), the linear probing and 1% semi-supervised performance would both incur a drop of 0.2%. Moreover, we further compare with FSM in terms of transfer few-shot learning and we find that FSM matching method achieves 68.5% and 80.2% in terms of 4-shot and 10-shot transfer few-shot accuracy on Pet37, respectively. In contrast, the proposed overlapping-based matching method significantly improves 4-shot and 10-shot transfer few-shot accuracy by 1.5% and 0.9%, respectively, demonstrating the effectiveness of the proposed matching method.

**The number of experts matters** We conducted an ablation study concerning the number of experts $n_e$, as detailed in Table 7. Our findings suggest that increasing the number of experts can lead to an increase in both linear evaluation performance and 1% few-shot performance. For instance, by increasing the experts' number from 2 to 16, the linear evaluation and 1% few-shot performance significantly increase by 1.3% and 5.6%, respectively.

Table 7: Comparison between different numbers of experts $n_e$ for the proposed CR-MoE. Linear probing (denoted as linear) and 1% imagenet semi-supervised (denoted as 1%) performance are reported. All the reported accuracy is top 1 accuracy (%).

| $n_e$ | Linear | 1% |
|---|---|---|
| 2 | 69.4 | 50.2 |
| 4 | 70.2 | 53.1 |
| 8 | 70.6 | 55.3 |
| 16 | **70.7** | **55.8** |

## 5 Conclusion

In this work, we study an efficient way of scaling contrastive learning with sparse Mixture of Experts. We start from naively plugging in the MoE to CL and observe that the naive combination tends to route different views of the same image to different subsets of experts, thus breaking invariant feature learning and hurting the performance of downstream tasks. To tackle this problem, we propose a novel regularization framework to promote consistency of experts selection on the same (or overlapped) image tokens while encouraging diversity of the experts selection for different images. Extensive evaluations on multiple downstream tasks demonstrate the proposed framework, CR-MoE, effectively improves the routing consistency and the overall performance of downstream tasks without increasing the computation cost.

**Broader Impact and Limitation** The proposed CR-MoE show the possibility of scaling Contrastive Learning with a large sparse neural network, which greatly reduces the training and inference time and energy consumption while achieving state-of-the-art performance. They can serve the goal of GreenAI for self-supervised learning. On the other hand, in this work, we mostly focus on academic datasets. However, in practice, unlabeled datasets in the wild may come with imbalances and adversarial samples, which could lead to performance or fairness issues. One of the future directions is to extend CR-MoE to such imbalance or adversarial settings.

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

## A Appendix

### A.1 With Other Backbones

Table 8: Illustration of the performance in terms of linear probing (denote as Linear), 1% semi-supervised performance (denote as 1%). Pet37@4-shot and Pet37@10-shot represent the transfer few-shot learning performance on Pet37 when 4 and 10 samples are available for each class, respectively. All the reported accuracy is top 1 accuracy (%).

| Match method | Model | Linear | 1% | Pet37@4-shot | Pet37@10-shot |
|---|---|---|---|---|---|
| Moco v3 Chen et al. (2021b) | ViT-B/16 | 76.7 | 63.9 | 74.2 | 84.5 |
| CR-MoE (ours) | V-MoE-B/16 | 76.3 | 64.9 | 76.7 | 85.0 |

In this section, we explore the performance of the proposed method on a different backbone. As shown in Table 8, CR-MoE with V-MoE-B/16 surpasses the Moco V3 with ViT-B/16 by 1% in terms of the 1% few-shot performance while leading to a small drop of 0.4% on linear evaluation performance. Moreover, it improves the transfer few-shot learning performance on Pet37 by 2.5% and 0.5% for 4-shot and 10-shot performance, respectively.

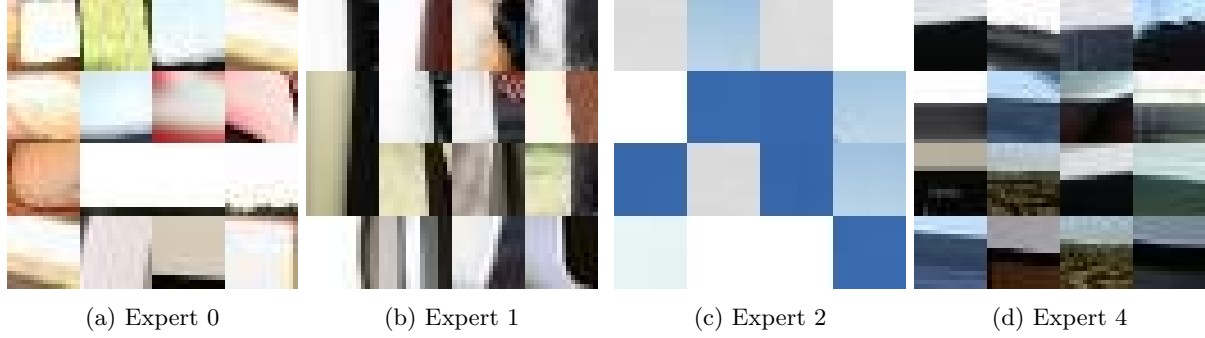

(a) Expert 0     (b) Expert 1     (c) Expert 2     (d) Expert 4

Figure 5: Visualization of the patch tokens routed to different experts in the **1st** layer of CR-MoE on ImageNet. The patches with different patterns are routed to different experts.

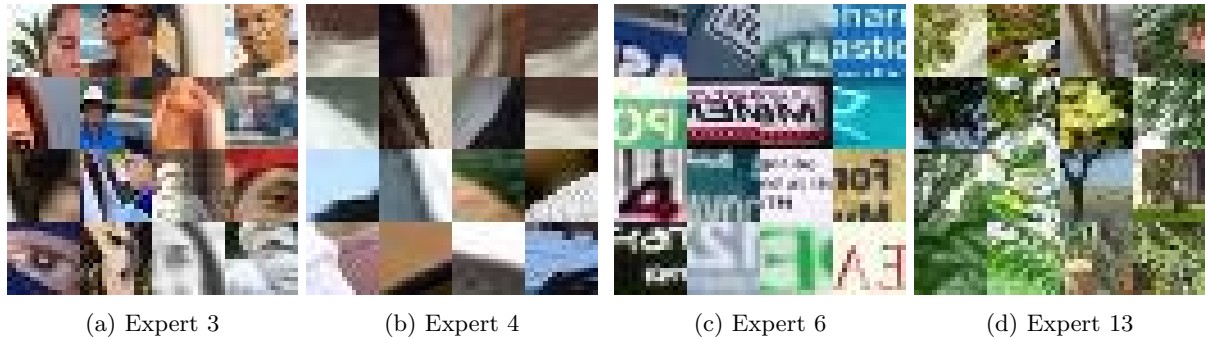

| (a) Expert 3 | (b) Expert 4 | (c) Expert 6 | (d) Expert 13 |

Figure 6: Visualization of the patch tokens routed to different experts in the **11th** layer of CR-MoE on ImageNet. The patches with different patterns are routed to different experts.

## A.2 Visualizing experts routing of more layers

We further analyze the routing pattern of 1st and 11th layers for CR-MoE, which correspond to the first and last MoE layers in the network, respectively. As shown in Figure 5 and Figure 6, similar patches would also be routed to the same experts on these layers. Moreover, we find that the patches with the same low-level pattern (e.g. edges) are often routed to the same expert in the shallow layer (e.g. the 1st layer). Meanwhile, the patches with similar semantic information are often routed to the same expert in the deep layer (e.g. the 7th and 11th layers).

