# OpenReview forum: "CR-MoE: Consistent Routed Mixture-of-Experts for Scaling Contrastive Learning"
_TMLR — Accepted by TMLR_

### Review · Reviewer_N628 · 2023-09-23

**Summary Of Contributions:**

This paper proposes to add a consistency loss in MoE-based visual encoder models for contrastive visual representation learning. MoE is a type of visual transformer model where each block has multiple parallel expert functions and a gating function that selects a distribution of experts. The problem with MoE is that the gating function can select different experts for the same image, and similar experts for different images, which makes the contrastive learning objective less effective. To mitigate the problem, the paper proposes to add a contrastive loss on the gating distribution. For classification tokens, it treats the gating function as an additional “feature,” and encourages the gating to be similar on positive pairs and dissimilar on negative pairs. For patch tokens, the local overlap IoU is taken into consideration when computing the loss on the gating functions. The resulting method show case that it can improve slightly upon regular vision transformers, especially on lower data regime and transfer learning settings. However, the differences do not seem too significant. It is not much better than standard contrastive learning, except on transfer learning tasks. It is not better than FSM. In conclusion, the paper may be technically correct but the results may not be significant.

**Audience:**

Yes

**Claims And Evidence:**

Yes

**Requested Changes:**

- Add large scale experiments
- Add hyperparameter studies
- Improve paper writing
- Fix citations

**Strengths And Weaknesses:**

**Strengths**
- The methodology of encouraging consistency between two gating functions on the global and local levels is well motivated.
- The proposed method shows a wide margin on transfer learning.

**Weaknesses**
- The MoE model is not really widely adopted, hence affecting the impact of the paper.
- The difference between regular ViT and regular MoE using MoCo is small. Table 2 last two rows of results show that the added loss is not really helping too much, so the only noticeable benefit is on transfer learning.
- The difference between the proposed method and the cited FSM method is non-existent (Table 6).
- The motivation of the paper is to scale models more efficiently using sparsely connected experts. The experiments do not showcase that. We don’t know if MoE with the proposed changes is really going to scale vision models as the results focus on relatively light weight models.
- There is no analysis on the alpha parameter. It would be good to have a sweep of hyperparameters to show the sensitivity and the necessity of each loss component.
- The paper writing has a large room to improve. In the introduction, it assumes knowledge of the MoE model with a lot of jargon, which is hard to parse before reading the background section. Figure 1 is not understandable until introducing the model section. E.g. what is corresponding token, what is shared expert, what is shared weight branch and independent branch. I would suggest moving it to experiments and properly defining the terminologies.
- All of the citations need to be fixed. Choose properly between \citep and \citet.

---

### Review · Reviewer_cRMt · 2023-10-01

**Summary Of Contributions:**

Scaling up the number of parameters for neural networks often brings perfomance gains, at the cost of increased compute budget.
The idea of "Mixture of Experts" (MoE) from Riquelme et al. 2021 addresses this issue, where there are multiple MLP layers in each transformer block  and only a few of them are selected to process each token by a gating mechanism.
This way, the number of parameters is increased (hence the complexity of the original network) while adding a marginal computational complexity overhead.

This paper proposes a method that allows training contrastive self-supervised learning (SSL) methods with more parameters while maintaining their compute requirements.
It is claimed that MoE cannot be applied to MoCo-v3 naively in this setting, because patches that are shared between the two images of a positive pair can be assigned to different "experts".
Therefore, the weight-sharing mechanism of contrastive SSL is not valid anymore.
To circument this problem, a new loss term is proposed that reqularizes the assignment of patch tokens across experts.
The experiments show that the proposed model trained on ImageNet-1K learn better representations when evaluated on ImageNet-1K or 3 transfer datasets (CIFAR10, Pet37, Food101).

**Audience:**

Yes

**Broader Impact Concerns:**

I do not see a need for a broader impact statement.

**Claims And Evidence:**

No

**Requested Changes:**

I expect the authors to respond to my concerns and improve the manuscript accordingly.

**Strengths And Weaknesses:**

### Strengths

It is often shown that self-supervised methods benefit from scaling more than supervised counterparts.
Proposing ways to train self-supervised methos in more efficient/effective ways is a valuable contribution to the field.
The proposed method with roughly 3x more parameters than the baseline (but with similar FLOPs) improves the baseline on downstream tasks.

### Weaknesses

According to the motivation of the paper, i.e. maintaining the consistency of gating tokens that belong to the same patch, I am not sure how Equations-6/7 help in that regard.
First of all, notation is not clear.
These equations follow the notation from Riquelme et al. 2021, but there is no need to specify tokens for different images.
Currently, Equation-6 is no different than Equation-2 (the original InfoNCE loss of MoCo-v3), yet Equations-2/6 both appear in the ultimate loss (Equation-10) used to train the model.
Similarly, Equation-7 just "filters" tokens which do not overlap with any other token according to some threshold $\lambda$.
Which means that, Equation-7 is actually a mechanism to do a better "pair selection".
Therefore, I'd strongly encourage the authors to compare to works like [A] or to baselines where MoCo-v3 is trained by positive pairs which overlap with at least some threshold.

Based on my previous comment, the illustrations shown in Figure-1/3 might be misleading.
As Equation-7 filters some patch tokens**$\dagger$, the number of expert selection is reduced in consequence.

Experiments do not extensively validate the main claim of the paper: scaling the number of parameters.
The proposed method is only trained with the ViT-S architecture containing 72M parameters (for 16 experts) and compared against ViT-S, with 22M parameters.
To validate the claims, it'd be nice to see:
- How the number of experts impact the results
- The performance of a network with the same number of parameters (72M)

The weight of the proposed loss $w_G$ is so small, i.e. 1e-3.
I wonder whether this is because the range of this loss term is high.
Otherwise it is surprising that the proposed loss actually improves performance.
This makes me question even more whether the improvements come from the careful selection of tokens for positive pairs (my comment above)

Visualization of the patch tokens routed to different experts in Figure-4 is obtained by the assignments of the CR-MoE layer in the 7th transformer block.
I wonder if such behaviour is generalized across the network, or Figure-4 is just cherry-picked.

[A] What Makes for Good Views for Contrastive Learning? Tian et al. 2020, NeurIPS 2020
Finds best positive pairs for self-supervised contrastive learning.

$\dagger$: It would also be good to show how many tokens are discarded based on different thresholds.


### Some minor points

- Figure-1 is misplaced, or its caption should be much more informative.
While reading the introduction first time, I had to check the method and experiments sections to actually understand this figure.
For instance, it wasn't clear what "expert" was, what "corresponding tokens" meant, or out of how many experts the y-axis ranged
Maybe Figure-3a can be moved next to Figure-1.
- What is FFN?
- applying similarity constrain --> applying THE similarity constrain

---

### Review · Reviewer_Fz5M · 2023-10-15

**Summary Of Contributions:**

This paper proposed Consistent Routed Mixture-of-Experts (CR-MoE) for self-supervised contrastive learning (CL). Based on the native combination of CL and MoE, this paper introduced consistent routing regularization, which encourages the corresponding tokens to be routed to the same set of experts.

**Audience:**

Yes

**Claims And Evidence:**

Yes

**Requested Changes:**

1. Modify the citation format.
2. Write the loss function.
3. Comparison with similar methods.
4. Comparison of the computation costs.

**Strengths And Weaknesses:**

Strengths:
1. The method is simple and effective. It may be an interesting paper for other researchers doing MoE.
2. The paper is well-written and easy to follow.
3. The experiments are clear and well-presented.

Weaknesses:
1. The citation is really bad. I think the authors should read the style template to distinguish between \citet{} and \citep{}.
2. I think the authors should write the loss function completely (Eq. 6 and 7) instead of just referencing it.
3. In Sec. 3.2, the authors reference two papers and say that those papers also proposed to enforce regional regularization. However, in the experiments, there are no comparisons against such methods.
4. As other vision MoE (e.g., vMoE) paper shows, after training, similar patches will automatically be routed to the same expert. Why such a phenomenon is different in CL, i.e., you need extra regularization to enforce similar patches being routed to the same expert?
5. How generalizable is this method? E.g., how about applying it to the supervised training?
6. I think the training and inference computation costs should also be compared. MoE approach should be slower in practice.

---

### Decision · Action_Editor_rGcy · 2023-12-23

**Recommendation:** Accept with minor revision

**Comment:**

This paper studies applying contrastive learning (CL) to Mixture of Expert (MoE) model, in order to address the scaling issue brought by training with large, dense models. The paper identifies that a naive MoE+CL approach tends to route two augmented views of the same image token to different subsets of experts, which contrasts with the CL working mechanism and affects invariant feature learning; the paper then proposes a regularization by enforcing expert-routing similarity between different views of the same image and promoting expert-routing diversity of patches from different images. Results show the efficacy of the proposed regularization.

Reviewer cRMt has doubts on the validity of Eqns (6) and (7), i.e., the proposed regularization,  and thinks that its efficacy for performance improvement is not fully tested; cRMt thinks that Eqns (6) and (7) function as "pair selection", rather than as regularization of the routing in MoE, and consequently the improvement may come from pair selection. AE reads the paper and thinks that cRMt may have misunderstanding of the paper - although  Eqns (6) and (7) look similar to Eqn. (2) in terms of the function form, their effect on learning is totally different; in fact they impose regularization on routing of experts, while Eqn. (2) of CL is for feature learning itself. Authors have made such clarifications in the response and revised paper. Another important suggestion from cRMt is to study the routing effects from the proposed regularization for transformer blocks other than the 7th one, for which the authors conducted further experiments and show similar routing effects across the ViT network.

Both reviewers cRMt and N628 point out that applying the proposed method on ViT-S cannot fully justify the proposed regularization; the authors conducted experiments on the larger ViT-B and got consistent results. Reviewer N628 worries about impact of the work since MoE models are not widely used, which however, does not affect decision making for TMLR. In the revised paper, the authors also added additional analysis on the alpha parameter, as required by N628.

Reviewer Fz5M suggests to compare the training and inference computation costs for the MoE version of ViT; the authors conducted the comparison in the revision and the results confirm the claims made in the paper. Reviewers also point out writing issues of the paper; most of these have been fixed in the revised paper, but there exist some remaining ones. For example, there is a typo in the denominator of Eqn. (6); in P5, spatially correlation -> spatial correlation.

Overall, the paper is technically correct but the results are not very significant. Given that two reviewers support acceptance of the paper, while the other reviewer has a misunderstanding of the paper, we recommend acceptance. The authors please thoroughly check the paper again and make sure there are no more typos. Please also make sure all the revisions made in the response phase are included in the final version.

**Audience:**

Yes

**Claims And Evidence:**

Yes

---

> ### Author Response · Authors · 2024-01-25
> **Thanks Action Editor**
>
> We extend our gratitude to the action editor for your diligent efforts. The camera-ready version of the document has been uploaded, and we have addressed your comments accordingly.

---

> > ### Comment · Editors_In_Chief · 2024-02-09
> >
> > Hey folks, the author block doesn't follow the TMLR format. Can you please correct it?
> >
> > Gautam

---

> > > ### Author Response · Authors · 2024-02-09
> > > **Author block corrected**
> > >
> > > We have corrected the author block.

---

> > > > ### Comment · Editors_In_Chief · 2024-02-10
> > > >
> > > > Thanks. Kui, can you please verify this at your earliest convenience?
> > > >
> > > > Thanks,
> > > > Gautam